

# Composite learning tracking control for underactuated marine surface vessels with output constraints

Huaran Yan[1], Yingjie Xiao[1] and Honghang Zhang[2]

[1] Merchant Marine College, Shanghai Maritime University, Shanghai, China
[2] Maritime College, Zhejiang Ocean University, Zhoushan, China

## ABSTRACT

In this paper, a composite learning control scheme was proposed for underactuated marine surface vessels (MSVs) subject to unknown dynamics, time-varying external disturbances and output constraints. Based on the line-of-sight (LOS) approach, the underactuation problem of the MSVs was addressed. To deal with the problem of output constraint, the barrier Lyapunov function-based method was utilized to ensure that the output error will never violate the constraint. The composite neural networks (NNs) are employed to approximate unknown dynamics. The prediction errors can be obtained using the serial-parallel estimation model (SPEM). Both the prediction errors and the tracking errors were employed to construct the NN weight updating. Using approximation information, the disturbance observers were designed to estimate unknown time-varying disturbances. The stability analysis via the Lyapunov approach indicates that all signals of unmanned marine surface vessels are uniformly ultimate boundedness. The simulation results verify the effectiveness of the proposed control scheme.

Corresponding author
Huaran Yan, hry455596621@126.com

# INTRODUCTION

In recent years, with the development of the marine economy, marine transport vehicles have gained much attention (*Shen et al., 2020*; *Yu, Guo & Yan, 2019*). Marine surface vehicles (MSVs) have been widely used in marine exploration, marine transportation, marine survey and other fields (*Liu et al., 2016*; *Shao et al., 2019*). To accomplish these tasks, the trajectory tracking control of MSVs plays a significant role. Due to the influence of the external environment, the kinetics of MSVs inevitably have unknown dynamics and unknown time-varying environmental disturbances.

In view of this, a series of control approaches have been utilized for control of MSVs, including neural network (NN) control (*Zhu et al., 2021*; *Li et al., 2015*), fuzzy logic system (FLS) control (*Peng, Wang & Wang, 2018*; *Wang, Sun & Er, 2018*), disturbance observer-based (DOB) control (*Guo & Zhang, 2020*; *Hu et al., 0000*), and the finite-time control (*Zhu, Ma & Hu, 2020*; *Wang, Pan & Su, 2019*; *Wang & Deng, 2020*). In *Zhu et al. (2021)*, *Li et al. (2015)*, *Peng, Wang & Wang (2018)*, *Wang, Sun & Er (2018)*, NNs

and FLSs are used to approximate the uncertain terms, such as unmodeled dynamics, unknown dynamics. In *Guo & Zhang (2020)*, *Hu et al. (0000)*, a DOB control approach was adopted to compensate compound uncertainty of parameter perturbations and unknown disturbances. In *Do (2016)* and *Ghommam & Saad (2018)*, the dynamic uncertainties of MSVs were dealt with by parameter adaptive technique and a backstepping design tool.

To address the underactuation problem of MSVs, several control methods are introduced, such as additional control method (*Do, 2010*; *Park, Kwon & Kim, 2017*; *Chen et al., 2020*), output redefinition control (*Shojaei & Arefi, 2015*; *Shojaei, 2017*), line-of-sight (LOS) (*Shojaei, 2015*; *Gao et al., 2016*; *Jia, Hu & Zhang, 2019*; *Liu, 2019*), etc. Three additional control terms were adopted to address the underactuation problem of MSV in *Do (2010)*, *Park, Kwon & Kim (2017)*, *Chen et al. (2020)*. To achieve the design of trajectory tracking control laws, the output redefinition control approach in *Shojaei & Arefi (2015)* and *Shojaei (2017)* was introduced to handle the underactuation problem, the combination of adaptive technique, NNs and saturation function to solve the unknown disturbances, unknown dynamic and input saturation, respectively. In *Shojaei (2015)*, *Gao et al. (2016)*, *Jia, Hu & Zhang (2019)* and *Liu (2019)*, the LOS method was utilized to solve the underactuation problem of MSVs, the combination of parameter adaptive technology and NN approximation are used to successfully solve the time-varying external disturbance and parameter uncertainty.

For the sake of navigation safety, the output constraint problem is inevitably in practice. In practice, the navigable water areas are restricted, and then surface vessels should navigate in the navigable water areas. When the position error is too large, it may lead to collision accident of MSVs. When the yaw angle errors become excessive, the actuator will be damaged due to overload. Therefore, it is necessary to further study the MSVs trajectory tracking system with output constraints. Several methods have been presented to solve the output constraint problem, such as moving-horizon optimal control (*Mayne & Michalska, 1990*), artificial potential field (*Sun & Ge, 2014*), barrier Lyapunov function (BLF) (*Tee et al., 2011*) and output error transformation method (*Zheng et al., 2020*; *Zhu, Du & Kao, 2020*). In *Zheng et al. (2020)* and *Zhu, Du & Kao (2020)*, the output constraint problem is transformed into a tracking error constraint problem by using the coordinate transformation. Coordinate transformation ensures that the tracking error always stays within predefined boundaries. Duo to the structure of Lyapunov function can be constructed by a barrier function, the BLF-based approach can solve the problem of trajectory tracking control for MSVs under the output constraint (*Zhu, Du & Kao, 2020*; *Zhao, He & Ge, 2014*). In simultaneous consideration of unknown dynamics and time-varying disturbances, *Zhu, Du & Kao (2020)* use a log-BLF method to solve the constant symmetric output constraint, *Zhao, He & Ge (2014)* utilize the asymmetric BLF method to deal with the asymmetric output constraints.

All the literature mentioned before have concentrated on the tracking and stability of the system. Most literature have not mentioned the precision accuracy of identifying models. In practice, the model uncertainty should be approximated as precisely as possible. In generally, the unknown dynamics of the system can be compensated by using adaptive control technique. In order to achieve better control performance, composite adaptive

control scheme is developed in *Patre & Bhasin (2010)*. It makes the system realize faster parameter convergence as well as smaller tracking error, and has been applied in various fields (*Sun, Pan & Yang, 2017*; *Pan, Sun & Yu, 2016*). By approximating the unknown dynamic items faster and more accurately to obtain better control performance, the prediction errors can be constructed by the serial-parallel estimation model (SPEM) (*Peng, Wang & Wang, 2017*). Then, the updating law of the neural network is designed by using the prediction error, which improves the transient performance effectively. To update the laws and optimize the system's transient performance, *Yucelen & Haddad (2013)* presented an adaptive control modification. An error feedback term was included in the reference model in *Pan, Sun & Yu (2016)* and *Stepanyan & Krishnakumar (2010)* to improve the transient performance of the model. In *Xu & Sun (2018)*, both the prediction errors and the tracking errors were applied to construct the updating law of NNs weights. The index of learning performance is introduced in the update rate, some literature focus on constructing composite learning laws by introducing auxiliary filter (*Na et al., 2015*; *Huang et al., 2018*) or using time interval data (*Xu et al., 2019*; *Xu et al., 2018*).

In this paper, we propose a composite learning control strategy for underactuated MSVs subject to unknown dynamics, ocean environmental disturbances, and output constraints based on the discussion above. The main contributions can be summarized as follows.

- Position error and yaw angle error constraints are addressed by employing the BLF-based method. The dynamic surface control approach is used to decrease the computation of the explosion problem that exists in the backstepping method.
- The composite NNs are employed to approximate the unknown dynamics of MSVs. Different from the traditional NN in which only the tracking errors are used to update the NN weights, both the tracking errors and prediction errors are used to update the NN weights. Therefore, the unknown dynamics can be approximated faster and more accurately.
- Using the approximation to the unknown dynamics of MSVs, the NDOs are constructed to estimate time-varying disturbances. By combining the dynamic surface control technique with disturbance observers and composite NNs, a trajectory tracking control system is developed. Compared with the control scheme based on neural networks, the proposed control scheme can effectively improve the transient and steady-state performance of MSVs trajectory tracking control.

The rest of this paper is arranged as follows. In Section 2, the mathematical model of MSVs and the problem formulation are introduced. In Section 3, the principle of intelligent approximation using NN is presented. In Section 4, proposes the details of controller design procedures. In Section 5, the simulation results are given to show the effectiveness of the controller. In Section 6, the entire work is summarized.

## PROBLEM FORMULATION AND PRELIMINARIES

### MSV kinematic and dynamic models

The mathematical model of underactuated MSVs with 3 degrees of freedom can be described as

$$\dot{x} = u\cos\varphi - v\sin\varphi \tag{1a}$$

$$\dot{y} = u\sin\varphi + v\cos\varphi \tag{1b}$$

$$\dot{\varphi} = r \tag{1c}$$

$$\dot{u} = \frac{1}{m_{11}}(m_{22}vr - d_{11}u + \tau_u + \Delta f_u + d_u) \tag{2a}$$

$$\dot{v} = \frac{1}{m_{22}}(-m_{11}ur - d_{22}v + \Delta f_v + d_v) \tag{2b}$$

$$\dot{r} = \frac{1}{m_{33}}\left[(m_{11} - m_{22})uv - d_{33}r + \tau_r + \Delta f_r + d_r\right] \tag{2c}$$

where $[x, y, \varphi]^{\mathrm{T}}$ denotes the position and heading angle in the inertial reference frame. $[u, v, r]^{\mathrm{T}}$ denotes surge, sway and angular velocity in the body-fixed frame. The $m_{ii}, i = 1, 2, 3$ represent the inertia including added mass. The $d_{ii}, i = 1, 2, 3$ stand for the hydrodynamic damping in surge, sway and yaw. The $d_j, j = u, v, r$ denote unknown environmental disturbances. $\Delta f_u$, $\Delta f_v$ and $\Delta f_r$ represent unknown dynamics of the MSVs. $\tau_u$ and $\tau_r$ are the control force and moment in the surge and yaw directions.

**Assumption 1:** The environmental disturbances $d_j$ are unknown bounded and there exists $|d_j| \leq \bar{d}_j, j = u, v, r, \bar{d}_j$ are unknown positive constants.

**Remark 1:** The ocean disturbances include slowly changing disturbances caused by second-order waves, currents, winds and unknown dynamics, as well as norm-bound disturbances caused by ocean uncertainties. The energy in the marine environment is finite. The rate of change of ocean disturbance is unknown bounded.

**Remark 2:** Since these parameters of MSVs are affected by operational conditions and marine environment. These factors change frequently, which makes these parameters of MSVs are uncertainties. where $m_{ii}$ and $d_{ii}, i = 1, 2, 3$ represent nominal values of the inertia including added mass and the hydrodynamic damping, respectively. Where $\Delta f_j, j = u, v, r$ represent unknown dynamics includes uncertain parts of the model parameters.

**Assumption 2:** The desired smooth reference signal $x_d$, $y_d$ and its first two time derivatives are bounded.

The position errors and orientation tracking errors will be defined in the body-fixed frame

$$x_e = (x - x_d)\cos\varphi + (y - y_d)\sin\varphi \tag{3a}$$

$$y_e = -(x - x_d)\sin\varphi + (y - y_d)\cos\varphi \tag{3b}$$

The time derivative of Eqs. (3a) and (3b) can be expressed as

$$\dot{x}_e = u + ry_e - \dot{x}_d\cos\varphi - \dot{y}_d\sin\varphi \tag{4a}$$

$$\dot{y}_e = v - rx_e + \dot{x}_d\sin\varphi - \dot{y}_d\cos\varphi \tag{4b}$$

In engineering practice, the MSV position, heading, velocities in surge and sway, and yaw rate can be measured by the global positioning system, the gyro compass, the Doppler log, and the rate gyro, respectively. Then, we define the tracking position error $\rho_s$ and yaw angle error $\theta$ as

$$\rho_s = \rho_e - \rho_0 = \sqrt{x_e^2 + y_e^2} - \rho_0 \tag{5a}$$

$$\theta = \arctan 2(y_e, x_e) \tag{5b}$$

By combining Eqs. (3a)–(3b) and Eqs. (5a)–(5b) we can get

$$x_e = \rho_e \cos\theta \tag{6a}$$

$$y_e = \rho_e \sin\theta. \tag{6b}$$

To avoid the possible singularity of the virtual control law, a positive constant $\rho_0$ is introduced. Considering Assumption 1 and Assumption 2, the control objective is to construct the composite intelligent learning control law $\tau_u$ and $\tau_r$ for MSVs to make sure the $\rho_s$ and $\theta$ can converge to arbitrarily small errors under unknown dynamics, time-varying disturbances and output constraints.

### Radial basis function neural network (RBFNN) approximation

In this paper, the RBF NNs are employed for approximation. For an arbitrary continuous function $f(\varsigma)$ over a compact set $\Omega(\varsigma) \to R^n$, there exists an RBF NN with the following form:

$$f(\varsigma) = \omega^T \psi(\varsigma) + \xi_w, \forall \varsigma \in \Omega(\varsigma) \tag{7a}$$

$$\psi(\varsigma) = exp(-(\varsigma - c_j)^T(\varsigma - c_j)/b_{jl}^2), j = 1, 2, \ldots, l \tag{7b}$$

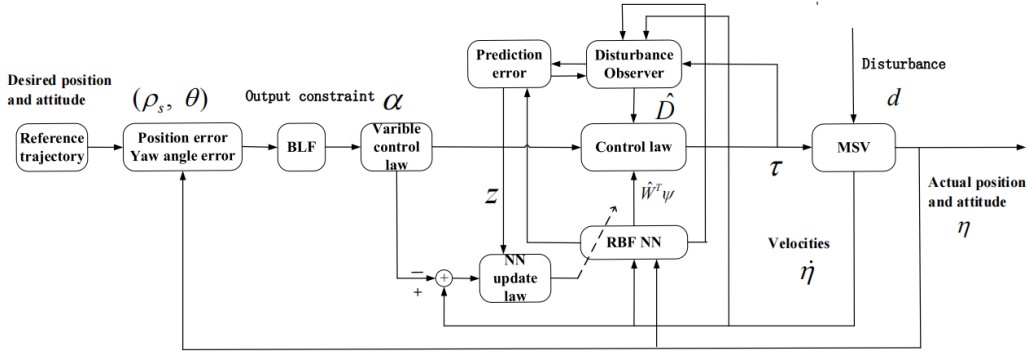

**Figure 1  Schematic of the MSV closed-loop tracking control.**

where $f(\varsigma) \in R^p$ denotes the output vector of the RBF NN, $\varsigma \in R^q$ denotes the input vector of the RBF NN. $\psi(\varsigma)$ is Gaussian basis function. $c_j$ is the center of the basis function and $b_j$ is the width of the Gaussian function. $\xi_w$ is the approximation error that satisfies $|\xi_w| \leq \bar{\xi}$, $\bar{\xi}$ is an unknown positive constant.

According to Eq.(43), $\omega$ is the ideal weight parameter that satisfies $\omega = \text{argmin}_{\omega \in R^\ell} \left\{ \sup_{\varsigma \in \Omega(\varsigma)} |f(\varsigma) - \omega^T \psi(\varsigma)| \right\}$ represent NN weights parameter. However, it is very difficult to determine the ideal weight parameter. $\hat{\omega}$ is the estimate of the NN weights parameter. However, it is very difficult to determine the ideal weight parameter. The estimate of the NN weights parameter is usually used to approximate the unknown nonlinear term such as $\hat{f} = \hat{\omega}^T \psi$ in practice.

## CONTROL LAW DESIGN

In this section, we can design the control law for the MSVs under Assumption 1–2. The block diagram of the trajectory tracking control system of MSVs is presented in Fig. 1. Combing Eqs. (5a) and (5b) with Eqs. (6a) and (6b), the time derivative of $\rho_s$ can be written as

$$\dot{\rho}_s = u\cos\theta + v\sin\theta + \cos\theta\zeta_1 + \sin\theta\zeta_2 \tag{8}$$

where $\zeta_1$ and $\zeta_2$ are defined as follows

$$\zeta_1 = -\dot{x}_d \cos\varphi - \dot{y}_d \sin\varphi \tag{9a}$$

$$\zeta_2 = \dot{x}_d \sin\varphi - \dot{y}_d \cos\varphi \tag{9b}$$

When MSV pass through a narrow passage, it is necessary to limit the position error $\rho_s$ to prevent vehicle collisions. The BLF can be selected as the following form

$$V_1 = \frac{1}{2}\log\frac{k_a^2}{k_a^2 - \rho_s^2} \tag{10}$$

where $log(*)$ is the natural logarithm of $(*)$, $k_a$ is the constraint of $\rho_s$, there exist $|\rho_s| < k_a$.

Taking time derivative of Eq. (10) , it can be further written as

$$\dot{V}_1 = \frac{\rho_s \dot{\rho}_s}{k_a^2 - \rho_s^2}$$

$$= \frac{\rho_s}{k_a^2 - \rho_s^2}(u\cos\theta + v\sin\theta + \cos\theta\zeta_1 + \sin\theta\zeta_2) \tag{11}$$

The virtual control law can be designed as

$$\alpha_u = \sec\theta(-k_\rho\rho_s - v\sin\theta - \cos\theta\zeta_1 - \sin\theta\zeta_2) \tag{12}$$

where $k_\rho$ is a positive constant.

In the surge direction, Let $\alpha_u$ pass through a first-order filter with a time constant $T_u > 0$ to get a new state variable $\beta_u$.

$$T_u \dot{\beta}_u + \beta_u = \alpha_u, \beta_u(0) = \alpha_u(0) \tag{13}$$

Then, the filter error and velocity error can be defined as $\lambda_u$ and $u_e$, respectively. So, it can be expressed as

$$\lambda_u = \beta_u - \alpha_u, u_e = u - \beta_u \tag{14}$$

The time derivative of $\lambda_u$ can be calculated as

$$\dot{\lambda}_u = -\frac{\lambda_u}{T_u} - \dot{\alpha}_u$$

$$= -\frac{\lambda_u}{T_u} + B_u \tag{15}$$

where $B_u$ is a continuous function and has a maximum value $H_u$.

Then, $V_2$ can be further chosen as

$$V_2 = \frac{1}{2}\log\frac{k_a^2}{k_a^2 - \rho_s^2} + \frac{1}{2}m_{11}u_e^2 \tag{16}$$

The time derivative of Eq. (16) can be written as

$$\dot{V}_2 = \frac{\rho_s \dot{\rho}_s}{k_a^2 - \rho_s^2} + m_{11}u_e\dot{u}_e$$

$$= \frac{\rho_s}{k_a^2 - \rho_s^2}(-u_e\cos\theta - \lambda_u\cos\theta - k_\rho\rho_s)$$

$$+ m_{11}u_e\dot{u}_e \tag{17}$$

According to Eqs. (2a) and (12), we can obtain the time derivative of as

$$m_{11}\dot{u}_e = m_{22}vr - d_{11}u + \tau_u$$

$$+\Delta f_u + d_u - m_{11}\dot{\beta}_u \qquad (18)$$

The unknown term can be approximate using NN. We have $m_{22}vr - d_{11}u + \Delta f_u = \omega_u^T\psi_u + \xi_u$. Here, let $D_u = \xi_u + d_u$. The $\xi_u$ is the approximation error that satisfies the time derivative of $\xi_u$ is bound. With Assumption 1, we can get

$$|D_u| \leq \chi_{u0}, |\dot{D}_u| \leq \chi_u \qquad (19)$$

where $\chi_{u0}$ and $\chi_u$ are unknown positive constants.

Therefore, the time derivative of $V_2$ can be further written as

$$\dot{V}_2 = \frac{\rho_s\dot{\rho}_s}{k_a^2 - \rho_s^2} + m_{11}u_e\dot{u}_e$$

$$= \frac{\rho_s}{k_a^2 - \rho_s^2}(u_e\cos\theta + \lambda_u\cos\theta - k_\rho\rho_s)$$
$$+ u_e(\omega_u^T\psi_u + D_u + \tau_u - m_{11}\dot{\beta}_u) \qquad (20)$$

Then, we can design the control law as

$$\tau_u = -\hat{\omega}_u^T\psi_u - \hat{D}_u + m_{11}\dot{\beta}_u - k_u u_e - \frac{\rho_s\cos\theta}{(k_a^2 - \rho_s^2)} \qquad (21)$$

where $k_u$ is a positive constant. $\hat{\omega}_u$ is the estimation of the $\omega_u$. $\hat{D}_u$ is the estimation of the $D_u$.

$$\tilde{\omega}_u = \omega_u - \hat{\omega}_u, \tilde{D}_u = D_u - \hat{D}_u \qquad (22)$$

From Eq. (21) along Eq. (20), we can get

$$\dot{V}_2 = \frac{\rho_s\lambda_u\cos\theta - k_\rho\rho_s^2}{k_a^2 - \rho_s^2} + u_e\tilde{\omega}_u^T\psi_u$$
$$+ u_e\tilde{D}_u - k_u u_e^2 \qquad (23)$$

Then, we can define $z_u$ as prediction error

$$z_u = u - \hat{u} \qquad (24)$$

$\hat{u}$ can be defined with SPEM

$$\dot{\hat{u}} = \frac{1}{m_{11}}(\tau_u + \hat{\omega}_u^T\psi_u + \hat{D}_u + \phi_u z_u) \qquad (25)$$

where $\hat{u}(0) = u(0)$, $\phi_u$ is a positive constant.

The prediction error is employed to construct the weight updating

$$\dot{\hat{\omega}}_u = \gamma_u[(u_e + \gamma_{zu}z_u)\psi_u - \vartheta_u\hat{\omega}_u] \qquad (26)$$

where $\gamma_u$, $\gamma_{zu}$ and $\vartheta_u$ are the positive constants to be designed.

The approximation information is employed to construct the NDO in the following form

$$\hat{D}_u = m_{11}u - \sigma_u \qquad (27a)$$

$$\dot{\sigma}_u = \hat{\omega}_u^T \psi_u + \hat{D}_u + \tau_u - (u_e + \gamma_{zu} z_u) \tag{27b}$$

According to Eqs. (2a), (27a) and (27b), the derivative of $\hat{D}_u$ can be expressed as

$$\dot{\hat{D}}_u = \tilde{\omega}_u^T \psi_u + \tilde{D}_u + u_e + \gamma_{zu} z_u \tag{28}$$

Then, the $\dot{\tilde{D}}_u$ can be calculated

$$\dot{\tilde{D}}_u = \dot{D}_u - \tilde{\omega}_u^T \psi_u - \tilde{D}_u - u_e - \gamma_{zu} z_u \tag{29}$$

Combining Eqs. (5a)–(5b) with Eqs. (6a)–(6b), the time derivative of $\theta$ can be written as

$$\dot{\theta} = -r + \frac{1}{\rho_e}(-u\sin\theta + v\cos\theta - \sin\theta\zeta_1 \\ + \cos\theta\zeta_2) \tag{30}$$

It is also necessary to restrict $\theta$ in practice, there exist $|\theta| < k_b$. Similar to the above, we select the following BLF candidates as

$$V_3 = \frac{1}{2}\log\frac{k_b^2}{k_b^2 - \theta^2} \tag{31}$$

Taking time derivative of Eq. (31), it can be further written as

$$\dot{V}_3 = \frac{\theta}{k_b^2 - \theta^2}(-r + \frac{1}{\rho_e}(-u\sin\theta + v\cos\theta \\ - \sin\theta\zeta_1 + \cos\theta\zeta_2)) \tag{32}$$

According to Eq. (32), we can get virtual control law $\alpha_r$ for the yaw direction

$$\alpha_r = k_\theta\theta + \frac{1}{\rho_e}(-u\sin\theta + v\cos\theta - \sin\theta\zeta_1 \\ + \cos\theta\zeta_2) \tag{33}$$

where $k_\theta$ is a positive constant.

**Remark 3:** From Eq. (33), it can be seen $\alpha_r$ is undefined when $\rho_e = 0$. The positive constant $\rho_0$ is designed to make $\rho_e - \rho_0$ can converge to the neighbor of zero. It means that $\rho_e$ can converge to the neighbor of $\rho_e$. Therefore, the singularity of $\alpha_r$ can be avoided.

Let $\alpha_r$ pass through a first-order filter with a time constant $T_r > 0$ to get a new state variable $\beta_r$.

$$T_r\dot{\beta}_r + \beta_r = \alpha_r, \beta_r(0) = \alpha_r(0) \tag{34}$$

Then, the filter error and velocity error can be defined as $\lambda_r$ and $r_e$, respectively. So, it can be expressed as

$$\lambda_r = \beta_r - \alpha_r, r_e = r - \beta_r \tag{35}$$

The time derivative of $\lambda_r$ can be calculated as

$$\dot{\lambda}_r = -\frac{\lambda_r}{T_r} - \dot{\alpha}_r$$

$$= -\frac{\lambda_r}{T_r} + B_r \tag{36}$$

where $B_r$ is a continuous function and has a maximum value $H_r$.

Then, $V_4$ can be further chosen as

$$V_4 = \frac{1}{2}\log\frac{k_b^2}{k_b^2 - \varphi_e^2} + \frac{1}{2}m_{33}r_e^2 \tag{37}$$

The time derivative of Eq. (37) can be written as

$$\dot{V}_4 = \frac{\theta\dot{\theta}}{k_b^2 - \theta^2} + m_{33}r_e\dot{r}_e$$

$$= \frac{\theta}{k_b^2 - \theta^2}(-r_e - \lambda_r - k_\theta\theta) + m_{33}r_e\dot{r}_e \tag{38}$$

According to Eqs. (2c) and (35), we can obtain the derivative of $r_e$ as

$$m_{33}\dot{r}_e = (m_{11} - m_{22})uv - d_{33}r + \tau_r + \Delta f_r$$
$$+ d_r - m_{33}\dot{\beta}_r \tag{39}$$

The unknown term can be approximate using NN. We have $(m_{11} - m_{22})uv - d_{33}r + \Delta f_r = \omega_r{}^T\psi_r + \xi_r$. we can define $D_r = \xi_r + d_r$, The $\xi_r$ is the approximation error that satisfies the time derivative of $\xi_r$ is bound. With Assumption 1, we can get

$$|D_r| \le \chi_{r0}, |\dot{D}_r| \le \chi_u \tag{40}$$

where $\chi_{r0}$ and $\chi_r$ are unknown positive constants.

Then, the time derivative of $V_4$ can be further written as

$$\dot{V}_4 = \frac{\theta}{k_b^2 - \theta^2}(-r_e - \lambda_r - k_\theta\theta)$$
$$+ r_e(\omega_r{}^T\psi_r + D_r + \tau_r - m_{33}\dot{\beta}_r) \tag{41}$$

Then, we can get

$$\tau_r = -\hat{\omega}_r^T\varphi_r - \hat{D}_r + m_{33}\dot{\beta}_r - k_r r_e + \frac{\theta}{k_b^2 - \theta^2} \tag{42}$$

where $k_r$ is a positive constant. $\hat{\omega}_r$ is the estimation of the $\omega_r$. $\hat{D}_r$ is the estimation of the $D_r$.

$$\tilde{\omega}_r = \omega_r - \hat{\omega}_r, \tilde{D}_r = D_r - \hat{D}_r \tag{43}$$

From Eqs. (41) along (40), we can get

$$\dot{V}_4 = \frac{\theta}{k_b^2 - \theta^2}(-\lambda_r - k_\theta\theta) + r_e\tilde{\omega}_r^T\psi_r + r_e\tilde{D}_r - k_r r_e{}^2 \tag{44}$$

Then, we can define $z_r$ as prediction error

$$z_r = r - \hat{r} \tag{45}$$

$\hat{r}$ can be defined with SPEM

$$\dot{\hat{r}} = \frac{1}{m_{33}}(\tau_r + \hat{\omega}_r^T \psi_r + \hat{D}_r + \phi_r z_r) \tag{46}$$

where $\hat{r}(0) = r(0)$, $\phi_r$ is a positive constant.

The prediction error is employed to construct the weight updating

$$\dot{\hat{\omega}}_r = \gamma_r[(r_e + \gamma_{zr} z_r)\psi_r - \vartheta_r \hat{\omega}_r] \tag{47}$$

where $\gamma_r$, $\gamma_{zr}$ and $\vartheta_r$ are the positive constants to be designed.

The approximation information is employed to construct the NDO in the following form

$$\hat{D}_r = m_{33}r - \sigma_r \tag{48a}$$

$$\dot{\sigma}_r = \hat{\omega}_r^T \psi_r + \hat{D}_r + \tau_r - (r_e + \gamma_{zr} z_r) \tag{48b}$$

According to Eqs. (2a), (48a) and (48b), the derivative of $\hat{D}_r$ can be expressed as

$$\dot{\hat{D}}_r = \tilde{\omega}_r^T \psi_r + \tilde{D}_r + r_e + \gamma_{zr} z_r \tag{49}$$

Then, the $\dot{\tilde{D}}_r$ can be calculated

$$\dot{\tilde{D}}_r = \dot{D}_r - \tilde{\omega}_r^T \psi_r - \tilde{D}_r - r_e - \gamma_{zr} z_r \tag{50}$$

**Remark 4:** From Eqs. (26) and (47), it can easily obtain the weight updating of composite NN is designed by employing tracking error and prediction error. The prediction error can provide extra information for learning NN weight updating. Thus, better tracking performance can be achieved.

**Remark 5:** In Eqs. (26) and (47), $\gamma_u$ and $\gamma_r$ are positive constants used to optimize the learning rate. The $\hat{\omega}_u$ and $\hat{\omega}_r$ mainly tuned by the prediction errors if and are chosen larger, while if $\gamma_{zu}$ and $\gamma_{zr}$ are chosen smaller, the $\hat{\omega}_u$ and $\hat{\omega}_r$ mainly tuned by the tracking errors.

The compound unknown terms consist of unknown dynamics and time-varying disturbances are expressed as $\sum_u$ and $\sum_r$.

$$m_{22}vr - d_{11}u + \Delta f_u + d_u = \Sigma_u \tag{51a}$$

$$(m_{11} - m_{22})uv - d_{33}r + \Delta f_r + \tau_{wr} = \Sigma_r \tag{51b}$$

**Remark 6:** The disturbance observer and neural network contain each other's information. If compound unknown terms can be perfect follow by $\hat{\omega}_u^T \psi_u + \hat{D}_u$ and $\hat{\omega}_r^T \psi_r + \hat{D}_r$, the system's estimation of unknown information can be more accurate. As a result, the objective of composite learning combining NN and NDO is accomplished.

**Remark 7:** Through trial and error, we first choose the appropriate design parameters $k_\rho$, $k_\theta$, $k_u$, and $k_r$ to ensure that the system is stable. Furthermore, we properly regulate the other design parameters $\gamma_u, \gamma_{zu}, \gamma_r, \gamma_{zr}, \vartheta_u, \vartheta_r, \phi_u$ and $\phi_r$ to get the satisfactory control performance. A large number of simulations in many cases show that the larger $k_\rho$, $k_{\theta,}$, $k_u$, $k_r$ , $\gamma_{zu}$, $\gamma_{zr}$, $\phi_u$ and $\phi_r$ are, the MSVs can obtain higher tracking accuracy.

**Theorem 1:** Considering the closed-loop system Eqs. (1a)–(1c) and Eqs.(2a)–(2c) with unknown dynamics, time-varying disturbances and output constraint under Assumption 1–Assumption 2, if virtual control law Eqs. (12), (33), control law Eqs. (21), (42), the NN updating laws Eqs. (26), (47) and NDOs Eqs. (27a)–(27b), Eqs. (48a)–(48b) are designed. It is guaranteed that all signals include in Eq. (52) are uniformly ultimately bounded (UUB).

**Proof:** Consider the following Lyapunov function

$$V = V_2 + V_4 + \frac{1}{2}(\frac{1}{\gamma_u}\tilde{\omega}_u^T\tilde{\omega}_u + \tilde{D}_u^2 + \lambda_u{}^2$$
$$+ m_{11}\gamma_{zu}z_u^2 + \frac{1}{\gamma_r}\tilde{\omega}_r^T\tilde{\omega}_r + \tilde{D}_r^2$$
$$+ \lambda_r{}^2 + m_{33}\gamma_{zr}z_r^2) \tag{52}$$

The time derivative of Eq. (52) can be calculated as

$$\dot{V} = \dot{V}_2 + \dot{V}_4 + \frac{1}{\gamma_u}\tilde{\omega}_u^T(-\dot{\hat{\omega}}_u) + \tilde{D}_u(-\dot{\hat{D}}_u)$$
$$+ m_{11}\gamma_{zu}z_u\dot{z}_u + \lambda_u\dot{\lambda}_u + \lambda_r\dot{\lambda}_r$$
$$+ \frac{1}{\gamma_r}\tilde{\omega}_r^T(-\dot{\hat{\omega}}_r) + \tilde{D}_r(-\dot{\hat{D}}_r) + m_{33}\gamma_{zr}z_r\dot{z}_r \tag{53}$$

In the view of Eqs. (15), (36) and Young's inequality, we can get

$$\lambda_u\dot{\lambda}_u \le -\frac{\lambda_u{}^2}{T_u} + \frac{1}{2\iota}\lambda_u{}^2 + 2\iota H_u{}^2 \tag{54}$$

$$\lambda_r\dot{\lambda}_r \le -\frac{\lambda_r{}^2}{T_r} + \frac{1}{2\iota}\lambda_r{}^2 + 2\iota H_r{}^2 \tag{55}$$

Using Eqs. (26) and (47), we have

$$\frac{1}{\gamma_u}\tilde{\omega}_u^T(-\dot{\hat{\omega}}_u) = -\tilde{\omega}_u^T[(u_e + \gamma_{zu}z_u)\psi_u - \vartheta_u\hat{\omega}_u] \tag{56}$$

$$\frac{1}{\gamma_r}\tilde{\omega}_r^T(-\dot{\hat{\omega}}_r) = -\tilde{\omega}_r^T[(r_e + \gamma_{zr}z_r)\psi_r - \vartheta_r\hat{\omega}_r] \tag{57}$$

From Eqs. (29) and (50), we have

$$\tilde{D}_u\dot{\tilde{D}}_u = \tilde{D}_u(\dot{D}_u - \tilde{\omega}_u^T\psi_u - \tilde{D}_u - u_e - \gamma_{zu}z_u) \tag{58}$$

$$\tilde{D}_r\dot{\tilde{D}}_r = \tilde{D}_r(\dot{D}_r - \tilde{\omega}_r^T\psi_r - \tilde{D}_r - r_e - \gamma_{zr}z_r) \tag{59}$$

Combining Eqs. (2a)–(2c), (24), (25), (45) with Eq. (46), we can get

$$m_{11}\gamma_{zu}z_u\dot{z}_u = \gamma_{zu}z_u\left(\tilde{\omega}_u^T\psi_u + \tilde{D}_u - \phi_u z_u\right) \tag{60}$$

$$m_{33}\gamma_{zr}z_r\dot{z}_r = \gamma_{zr}z_r\left(\tilde{\omega}_r^T\psi_r + \tilde{D}_r - \phi_r z_r\right) \tag{61}$$

Combining Eqs. (23), (44), (52)–(58) and Young's inequality, Eq. (53) can be expressed as

$$
\begin{aligned}
\dot{V} \leq \quad & -(k_\rho - \frac{A}{2})\frac{\rho_s^2}{k_a^2 - \rho_s^2} - k_u u_e^2 - \gamma_{zu}\phi_u z_u^2 \\
& -\tilde{D}_u^2 - (\frac{1}{T_u} - \frac{1}{2A(k_a^2 - \rho_s^2)} - \frac{1}{2\iota})\lambda_u^2 \\
& +2\iota H_u^2 + \tilde{\omega}_u^T\vartheta_u\hat{\omega}_u + \tilde{D}_u\dot{D}_u - \tilde{D}_u\tilde{\omega}_u^T\psi_u \\
& -(k_\theta - \frac{1}{2})\frac{\theta^2}{k_b^2 - \theta^2} - k_r r_e^2 - \tilde{D}_r^2 \\
& -(\frac{1}{T_r} - \frac{1}{2(k_b^2 - \theta^2)})\lambda_r^2 - \gamma_{zr}\phi_r z_r^2 + \frac{1}{2\iota}\lambda_r^2 \\
& +2\iota H_r^2 + \tilde{\omega}_r^T\vartheta_r\hat{\omega}_r + \tilde{D}_r\dot{D}_r - \tilde{D}_r\tilde{\omega}_r^T\psi_r
\end{aligned}
\tag{62}
$$

According to Young's inequality, we can obtain

$$-\tilde{D}_g\tilde{\omega}_g^T\psi_g \leq \frac{1}{2}\zeta_g\tilde{D}_g^2\varpi_g^2 + \frac{1}{2\zeta_g}\tilde{\omega}_g^T\tilde{\omega}_g \tag{63}$$

$$\tilde{D}_g\dot{D}_g \leq \frac{1}{2}\tilde{D}_g^2 + \frac{1}{2}\chi_g^2 \tag{64}$$

$$\tilde{\omega}_g^T\hat{\omega}_g \leq -\frac{1}{2}\tilde{\omega}_g^T\tilde{\omega}_g + \frac{1}{2}\|\omega_g{}^*\|^2 \tag{65}$$

where $\zeta_g$ is positive user-defined parameter, $\|\psi_g\| \leq \varpi_g$, $\left|\dot{D}_g\right| \leq \chi_g$, $g = u, r$. $\chi_g$ and $\|\omega_g{}^*\|$ are positive constants.

From Eqs. (63)–(65), Eq. (62) can be expressed as

$$
\begin{aligned}
\dot{V} \leq \quad & -(k_\rho - \frac{A}{2})\frac{\rho_s^2}{k_a^2 - \rho_s^2} - k_u u_e^2 - (\frac{1}{2}\vartheta_u - \frac{1}{2\mu_u})\tilde{\omega}_u^T\omega_u \\
& -(\frac{1}{T_u} - \frac{1}{2A(k_a^2 - \rho_s^2)} - \frac{1}{2\iota})\lambda_u^2 - (\frac{1}{2} - \frac{1}{2}\mu_u\varpi_u^2)\tilde{D}_u^2 \\
& -\gamma_{zu}\phi_u z_u^2 - (k_\theta - \frac{1}{2})\frac{\theta^2}{k_b^2 - \theta^2} - k_r r_e^2 - \gamma_{zr}\phi_r z_r^2 \\
& -(\frac{1}{T_r} - \frac{1}{2(k_b^2 - \theta^2)} - \frac{1}{2\iota})\lambda_r^2 - (\frac{1}{2}\vartheta_r - \frac{1}{2\mu_r})\tilde{\omega}_r^T\omega_r \\
& -(\frac{1}{2} - \frac{1}{2}\mu_r\varpi_r^2)\tilde{D}_r^2 + 2\iota H_u^2 + \frac{1}{2}\vartheta_u\|\omega_u\|^2 + \frac{1}{2}\chi_u^2
\end{aligned}
$$

$$+2\iota H_r{}^2+\frac{1}{2}\vartheta_r\|\omega_r\|^2+\frac{1}{2}\chi_r^2$$

$$\leq -2aV+b \tag{66}$$

where $a=\min((k_\rho-\frac{A}{2}),k_u,(\frac{1}{T_u}-\frac{1}{2A(k_a^2-\rho_s^2)}-\frac{1}{2\iota}),\gamma_{zu}\phi_u,(\frac{1}{2}\vartheta_u-\frac{1}{2\mu_u}),(\frac{1}{2}-\frac{1}{2}\mu_u\varpi_u^2),(k_\theta-\frac{1}{2}),k_r,(\frac{1}{T_r}-\frac{1}{2(k_b^2-\theta^2)}-\frac{1}{2\iota}),(\frac{1}{2}\vartheta_r-\frac{1}{2\mu_r}),(\frac{1}{2}-\frac{1}{2}\mu_r\varpi_r^2),\gamma_{zr}\phi_r)$ , $b=2\iota H_u^2+\frac{1}{2}\vartheta_u\|\omega_u\|^2+\frac{1}{2}\chi_u^2+2\iota H_r{}^2+\frac{1}{2}\vartheta_r\|\omega_r\|^2+\frac{1}{2}\chi_r^2$.

By choosing the appropriate design parameters to make $k_\rho>\frac{A}{2},k_u>0,(\frac{1}{T_u}-\frac{1}{2A(k_a^2-\rho_s^2)}-\frac{1}{2\iota})>0,\gamma_{zu}\phi_u>0,(\frac{1}{2}\vartheta_u-\frac{1}{2\mu_u})>0,(\frac{1}{2}-\frac{1}{2}\mu_u\varpi_u^2)>0,k_\theta>\frac{1}{2},k_r>0,(\frac{1}{T_r}-\frac{1}{2(k_b^2-\theta^2)}-\frac{1}{2\iota})>0,(\frac{1}{2}\vartheta_r-\frac{1}{2\mu_r})>0,(\frac{1}{2}-\frac{1}{2}\mu_r\varpi_r^2)>0,\gamma_{zr}\phi_r>0$.

By solving Eq. (66), we have

$$0\leq V\leq \frac{b}{2a}+[V(0)-\frac{b}{2a}]e^{-2at} \tag{67}$$

From Eq. (67), we can obtain that $V\to\frac{b}{2a}$ as $t\to\infty$. All signals in the Lyapunov function Eq. (52) are UUB. This concludes the proof.

## SIMULATION RESULTS

In this section, to demonstrate the effectiveness of the proposed control system, the dynamic model of an MSV in *Do, Jiang & Pan (2004)* is considered.

The model parameters of the MSV are presented as follows: $m_{11}=120\times10^3\ kg,m_{22}=177.9\times10^3\ kg,m_{33}=636\times10^5\ kg\ m^2.\ d_{11}=215\times10^2\ kg/s,d_{22}=147\times10^3\ kg/s,d_{33}=802\times10^4\ kg/m^2s$.

The proposed control scheme is marked as $\tau_{CL}$. The control strategy without considering the prediction error is denoted as $\tau_{NN}$.

Case 1: The reference trajectory is selected as $x_d=200\sin(0.02t),y_d=200\cos(0.02t)$.

Unknown dynamics are selected as $[\Delta f_u,\Delta f_v,\Delta f_r]^T=[(-0.2d_{11}|u|)u,(-0.2d_{22}|v|)v,(-0.2|r|)r]^T$. The external disturbances are given as $[d_u,d_v,d_r]^T=[10^4\sin(0.3t-\pi/4)+10^4cos(0.2t+\pi/4)+2\times10^4,10^3\sin(0.2t-\pi/4)+10^3\cos(0.3t-\pi/4)+3\times10^3,10^5\sin(0.2t+\pi/6)+10^5\cos(0.5t-\pi/4)\ (-3\times10^5]^T$.

The initial condition is chosen as $[x(0),y(0),\varphi(0),u(0),v(0),r(0)]=[20,190,-0.02\pi,0,0,0]$. The control laws design parameters are designed as $\rho_0=10,k_\rho=0.4,k_u=6\times10^3,k_r=3.18\times10^6,T_u=0.8,T_r=0.3,\gamma_u=10000,\gamma_r=100,\gamma_{zu}=20,\gamma_{zr}=3000,\vartheta_u=0.00001,\vartheta_r=0.0001,\phi_u=10,\phi_r=1$.

Figures 2A–2F illustrate the simulation results for the MSV under the two control strategies. Fig. 2A clearly illustrates that the MSV can track the reference trajectory in the presence of unknown dynamics, time-varying disturbances and output constraint under two control methods. The result in Fig. 2B shows that MSV can accomplish faster and more precise tracking under $\tau_{CL}$. The results of approximation of unknown information in Figs. 2C and 2D further support this conclusion. The estimates of 2-norms weights are more sensitive under as illustrated in Fig. 2E. The control inputs $\tau_u$ and $\tau_r$ are plotted in Fig. 2F.

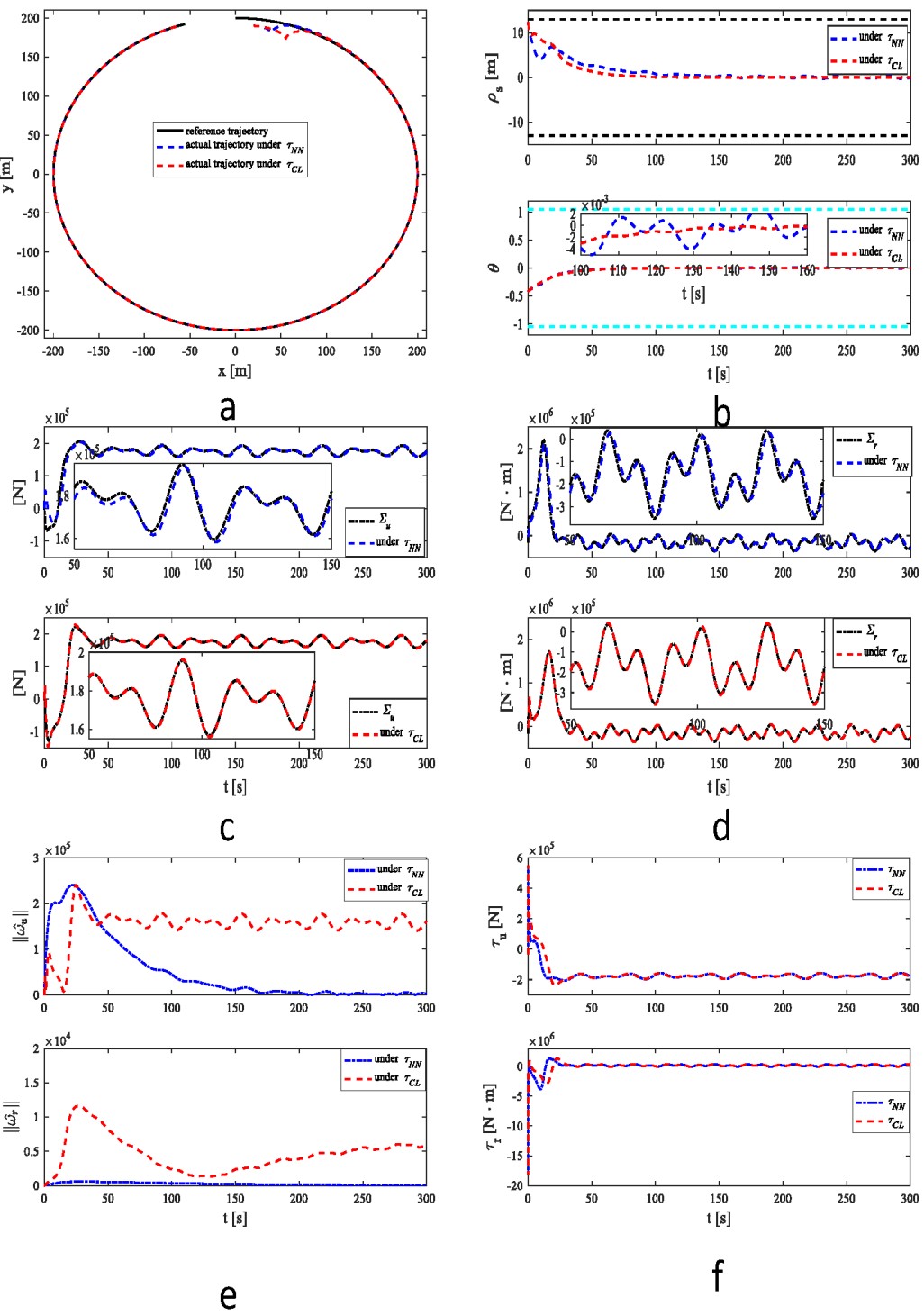

**Figure 2 Simulation results under $\tau_{NN}$ and $\tau_{CL}$ for case 1.** (A) Reference and actual trajectories of the MSV. (B) Tracking position error and yaw angle error. (C) $\sum_u$ and its estimation. (D) $\sum_r$ and its estimation. (E) 2-norms $\|\hat{\omega}_u\|$, $\|\hat{\omega}_r\|$ of parameter estimates $\hat{\omega}_u$ and $\hat{\omega}_r$. (F) Control signals $\tau_u$ and $\tau_r$.

Case 2: The MSV's unknown dynamics are raised $1.2 \times \Delta f_n$. The control law's initial conditions and design parameters are the same as in Case 1, and the larger time-varying disturbances can be chosen as $[d_u, d_v, d_r]^T = [1.5 \times 10^4 \sin(0.3t - \pi/4) + 1.5 \times 10^4 \cos(0.2t + \pi/4) + 3 \times 10^4, 1.5 \times 10^3 \sin(0.2t - \pi/4) + 1.5 \times 10^3 \cos(0.3t - \pi/4) + 3 \times 10^3, 1.5 \times 10^5 \sin(0.2t + \pi/6) + 1.5 \times 10^5 \cos(0.5t - \pi/4) - 4.5 \times 10^5]^T$.

Under two control systems, MSV can track a reference trajectory in the presence of unknown dynamics, time-varying disturbances and output constraint as shown in Fig. 3A. As demonstrated in Fig. 3B, MSV can obtain higher tracking performance under $\tau_{CL}$. The proposed control scheme has better robustness performance. As shown in Figs. 3C–3D, a similar result can be illustrated in case 1. The estimates of 2-norms weights are more sensitive under as illustrated in Fig. 3E. The control inputs are presented in Fig. 3F.

Case 3: The initial conditions and design parameters of the control law are the same as those in case 1. To further verify the superiority and effectiveness of the control scheme, another form of environmental disturbance are given as $[d_u, d_v, d_r]^T = d + h$. where $d$ is $d = [10^4 \sin(0.3t - \pi/4) + 10^4 \cos(0.2t + \pi/4) + 2 \times 10^4, 10^3 \sin(0.2t - \pi/4) + 10^3 \cos(0.3t - \pi/4) + 3 \times 10^3, 10^5 \sin(0.2t + \pi/6) + 10^5 \cos(0.5t - \pi/4) - 3 \times 10^5]^T$. $h$ is selected by the first-order Markov process. $\dot{h} = -\Lambda h + \Gamma \wp$, where $\wp \in R^3$ is the zero-mean Gaussian white noise.

The simulation results are depicted in Figs. 4A–4F. Under two control systems, MSV can track a reference trajectory under unknown dynamics, time-varying disturbances and output constraint as shown in Fig. 4A. As demonstrated in Fig. 4B, MSV can achieve better tracking performance under $\tau_{CL}$. As shown in Figs. 4C–4D, a similar result can be verified. The estimates of 2-norms weights are more sensitive under as shown in Fig. 4E. The control inputs are presented in Fig. 4F.

## CONCLUSIONS

In this paper, a composite learning trajectory tracking control scheme is proposed for underactuated MSVs in the presence of unknown dynamics, time-varying disturbances and output constraints. The underactuation problem of the MSVs is addressed by the LOS approach. The barrier Lyapunov function is introduced to deal with the problem of output constraint. The composite learning control scheme is utilized to approximate unknown dynamics. The prediction errors and the tracking errors are adopted to construct the NN weight updating. Using approximation information, the disturbance observers are designed to estimates unknown time-varying disturbances. The Lyapunov method is used to demonstrate the stability of a closed-loop system. The simulation results demonstrate the effectiveness and superiority of the proposed control scheme.

Furthermore, the finite-time control can be further considered. The control scheme in this paper can be easily combined with event-triggered control.

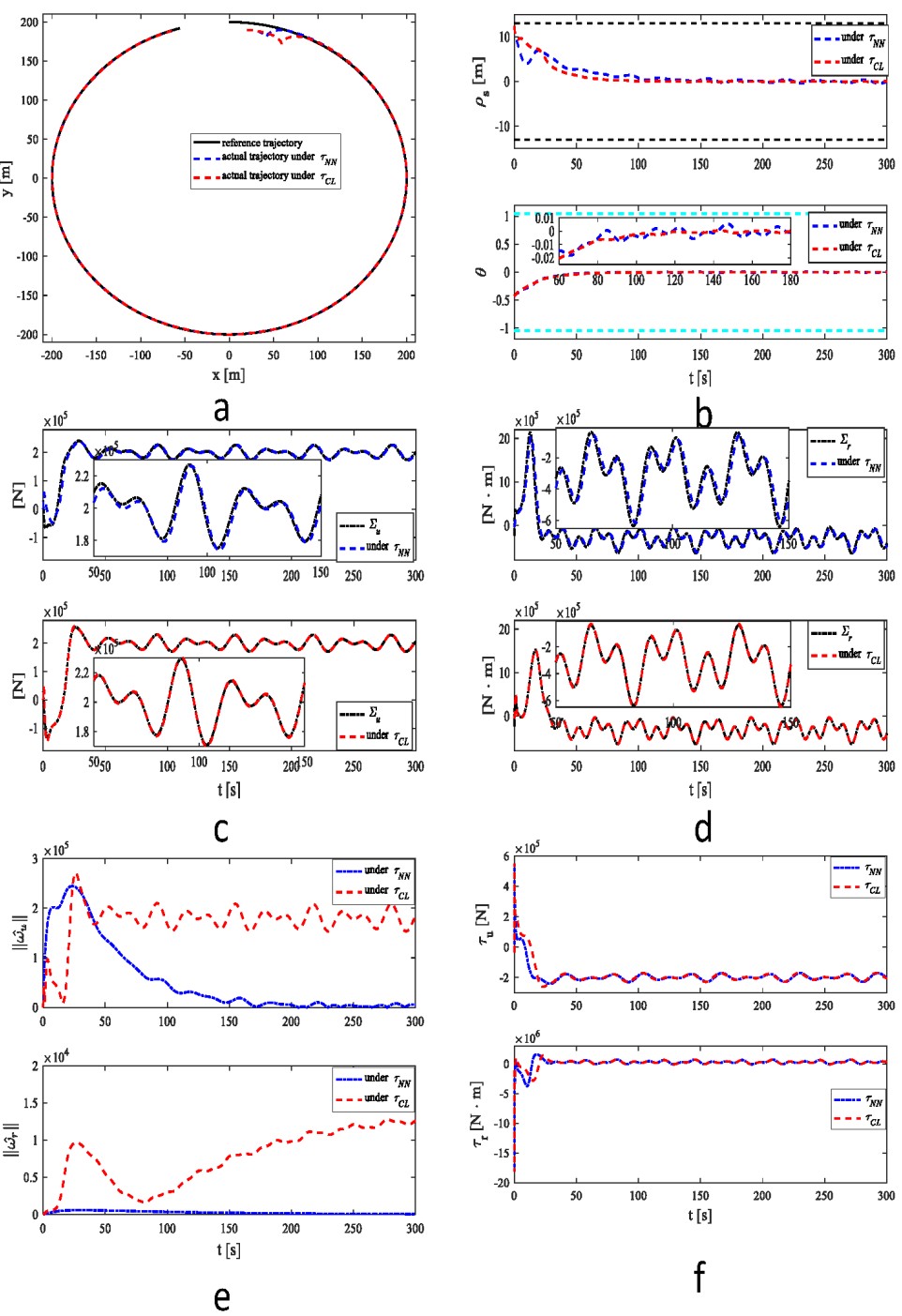

**Figure 3 Simulation results under $\tau_{NN}$ and $\tau_{CL}$ for case 2.** (A) Reference and actual trajectories of the MSV. (B) Tracking position error and yaw angle error. (C) $\sum_u$ and its estimation. (D) $\sum_r$ and its estimation. (E) 2-norms $\|\hat{\omega}_u\|$, $\|\hat{\omega}_r\|$ of parameter estimates $\hat{\omega}_u$ and $\hat{\omega}_r$. (F) Control signals $\tau_u$ and $\tau_r$.

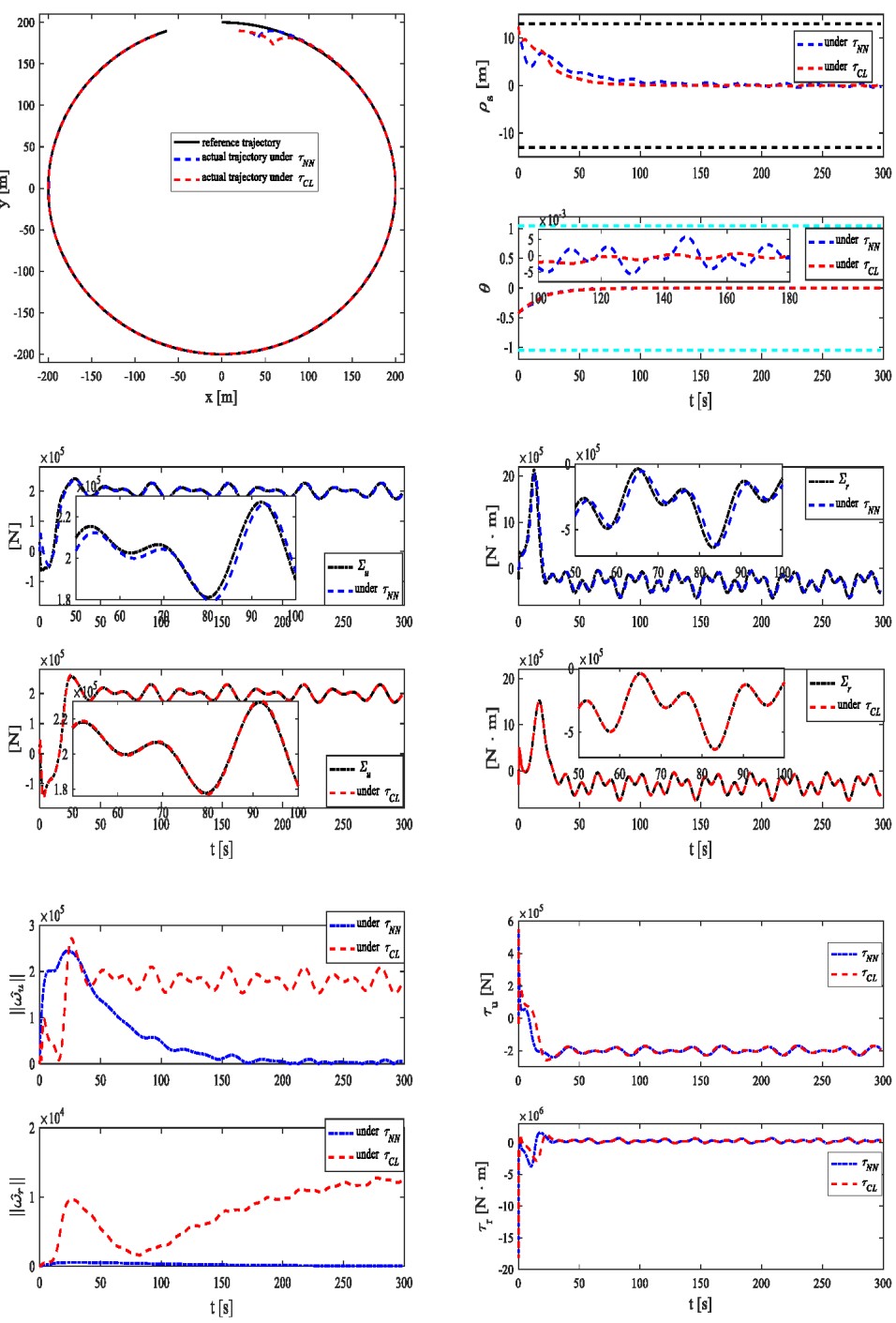

**Figure 4  Simulation results under $\tau_{NN}$ and $\tau_{CL}$ for case 3.** (A) Reference and actual trajectories of the MSV. (B) Tracking position error and yaw angle error. (C) $\sum_u$ and its estimation. (D) $\sum_r$ and its estimation. (E) 2-norms $\|\hat{\omega}_u\|$, $\|\hat{\omega}_r\|$ of parameter estimates $\hat{\omega}_u$ and $\hat{\omega}_r$. (F) Control signals $\tau_u$ and $\tau_r$.

### Funding

This work was supported by the National Natural Science Foundation of China under Grant 52071201. The funders had no role in study design, data collection and analysis, decision to publish, or preparation of the manuscript.

### Grant Disclosures

The following grant information was disclosed by the authors:
The National Natural Science Foundation of China: 52071201.

### Competing Interests

The authors declare there are no competing interests.

### Author Contributions

- Huaran Yan and Honghang Zhang conceived and designed the experiments, performed the experiments, analyzed the data, performed the computation work, prepared figures and/or tables, authored or reviewed drafts of the paper, and approved the final draft.
- Yingjie Xiao conceived and designed the experiments, analyzed the data, prepared figures and/or tables, and approved the final draft.

### Data Availability

The raw data is included in the article.

### Supplemental Information

Supplemental information for this article can be found online at http://dx.doi.org/10.7717/peerj-cs.863#supplemental-information.

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
