# Peer review of "Composite learning tracking control for underactuated marine surface vessels with output constraints"

_PeerJ Computer Science, doi:10.7717/peerj-cs.863_

## Round 0.1 · original submission · Minor Revisions

The paper is well-written with interesting topic. However some minor issues still exist and the quality of paper should be improved. For example, the expression, diagram, parameter selection, etc. Please follow the comments from the reviewers and proofread the re-submission carefully.

·

Basic reporting

This article solves the composite learning tracking control for the underactuated marine surface vessels. In particular, the output constraints are addressed in the control design procedure. The under-actuated ship motion control is a challenging problem. This article is well written and intersing. The composite learing technique is employed such that the control system has the learning capability. It can be considiered to be accepted after some revisions.

Somme comments are given as follows:
1. It seems that the mass parameters m_jj should be known. But, some other parameters are unknown, Some remarks are needed for the parameter determine.

2. The output constraints should be explained frome a practical viewpoint.

3. The ocean disturbances acting on the ships are assumed to be bounded and then are attenuated with the NN approximation errors. It has some conservativenss. Some disturbance estimtion scheme has been provided such as 10.1109/TITS.2021.3054177. Authors should be mentioned this issue in the context or lie in the future research.

Experimental design

No

Validity of the findings

No

Additional comments

No

Reviewer 2 ·

Basic reporting

This paper develops a composite learning trajectory tracking control scheme for underactuated marine surface vehicles (MSVs) with unknown dynamics and time-varying disturbances under output constraints. The considered issue is interesting and the paper has an acceptable structure and presentation. However, the following comments are needed to be considered in the revision:
1.It is suggested to supplement these more related works of composite adaptive, the advantages of composite learning control scheme can be clearly understood by readers.
2.How to choose the design parameters of the controller selected, in the simulation? Please give some remarks about it.
3.Please add some explanations of how the MSV obtains its positions, orientations, velocities and angular velocities..
4.It is suggested to supplement a block diagram of the control architecture

Experimental design

1. An extensive comparative study may help the authors to improve the paper presentation.
2. The format of Figures should be unified.

Validity of the findings

The trajectory tracking control for underactuated marine surface vehicles subject to unknown dynamics and time-varying disturbances under output constraints is considered. The problem investigated is practical and valuable.

---

## Round 0.2 · accepted · Accept

All the comments have been addressed well in the revised version and the reviewers satisfy the quality of the manuscript. Based on the contribution, I would like to recommend this manuscript.

·

Basic reporting

NA

Experimental design

NA

Validity of the findings

NA

Additional comments

NA

Reviewer 2 ·

Basic reporting

The article meets the PeerJ criteria and should be accepted as is.

Experimental design

The article meets the PeerJ criteria and should be accepted as is.

Validity of the findings

The article meets the PeerJ criteria and should be accepted as is.